# Dysbiosis-associated gut bacterium *Ruminococcus gnavus* varies at the strain level in its ability to utilize key mucin component sialic acid

Olga M. Sokolovskaya,[1] Jasmina Uzunovic,[2] Yutian Peng,[1] Mikiko Okumura,[3] Lingjue Mike Wang,[4] Yuhui Zhou,[3] Zijuan Lai,[4] Elizabeth Skippington,[2] Man-Wah Tan[1]

**ABSTRACT**  *Ruminococcus gnavus* is a prevalent human gut commensal bacterium with known roles in intestinal mucus degradation, including by catabolism of the terminal mucin sugar sialic acid. While *R. gnavus* is not considered a pathogen, overabundance of this species is correlated with inflammatory bowel disease (IBD), and its sialic acid metabolism may play a role in the dysbiotic state. Interestingly, liberation of mucin-bound sialic acid by *R. gnavus* yields the distinct product of 2,7-anhydro-N-acetylneuraminic acid (2,7-anhydro-Neu5Ac), in contrast to other known mucin-degrading bacteria, which generate Neu5Ac. This prompted us to look for 2,7-anhydro-Neu5Ac metabolism proteins in the genomes of 77 *R. gnavus* clinical isolates. We found that 2,7-anhydro-Neu5Ac metabolism is sporadically distributed in this species with respect to phylogeny and strain origin. We measured sialic acid-dependent growth of 12 sequenced isolates, finding that the presence of 2,7-anhydro-Neu5Ac catabolism proteins was predictive of growth on this substrate. Our analysis also uncovered "partial" 2,7-anhydro-Neu5Ac catabolism pathways in two *R. gnavus* strains, leading to the discovery that these strains grow with Neu5Ac as a sole carbon source, a metabolic capability previously unreported in this species. These results reveal a notable diversity of sialic acid catabolism across the *R. gnavus* species, an essential consideration for further investigations into the importance of this metabolism in mucin degradation and in the roles of *R. gnavus* in IBD and other gut dysbioses.

**IMPORTANCE**  *Ruminococcus gnavus* is a common resident of the human gut, but often blooms in inflammatory bowel disease (IBD). It remains unclear whether *R. gnavus* plays any direct role in disease occurrence. This study investigates *R. gnavus* utilization of sialic acid, a crucial component of host mucins, the main constituents of the mucus that lines and protects the gut. We profile variations in sialic acid metabolic pathways and growth among 77 *R. gnavus* strains, isolated from healthy people and individuals with IBD, discovering further diversity in sialic acid utilization than was previously appreciated. This study emphasizes that the intricate interaction between bacteria and their environment must be examined at the strain level to understand the roles of particular species in health and disease.

**KEYWORDS**  *Ruminococcus gnavus*, sialic acid, 2,7-anhydro-Neu5Ac, strain-level diversity, mucin glycans

*R*uminococcus (Mediterraneibacter) gnavus (1) is a commensal human gut bacterium that has come into the spotlight in the field of microbiomes due to its association with disease. While it is a prevalent gut bacterium, found in over 90% of individuals, multiple studies link *R. gnavus* overabundance with inflammatory bowel disease (IBD)

Address correspondence to Olga M. Sokolovskaya, olga.sokolovskaya.phd@gmail.com.

The authors declare no conflict of interest.

10.1128/spectrum.03090-24 **1**

(2–12). Despite reported associations, it remains unknown whether *R. gnavus* becomes abundant as a result of inflammation [for example, by higher tolerance of increased oxygen concentrations associated with the inflamed state, compared to other anaerobic gut bacteria (2)] or whether it exacerbates or plays causative roles in disease [for example, by surface presentation of "superantigens" (13), capsular polysaccharide (14), or production of pro-inflammatory factors (15)]. Importantly, the *R. gnavus* species contains significant strain-level variation, according to a pangenome analysis of 17 *R. gnavus* strains that reports 1,708–2,473 variable genes per genome, compared to a core genome of 1,178 genes (2). The strain-level variation in this organism, combined with the fact that individuals can harbor multiple strains, complicates questions about the potential roles of *R. gnavus* in health and disease.

A genomic analysis of *R. gnavus* strains published by Hall et al. reports that a sialic acid transporter is among 199 proteins found exclusively within a clade of *R. gnavus* that comprises IBD-associated strains, suggesting the relevance of sialic acid metabolism in the diseased state (2). In the colon, the region of the gut that bears the highest microbial load, sialic acids, particularly *N*-acetylneuraminic acid (Neu5Ac), are abundant terminal sugars on glycosylated mucins, the main constituents of the mucus lining of the gut (16). Because it caps glycan chains, Neu5Ac plays a role in mucus integrity, protecting mucins from degradation by bacterial glycoside hydrolases and mucinases (17). This role of Neu5Ac is notable as mucus degradation is a hallmark of IBD (18–22). Simultaneously, specialized mucin-degrading microorganisms express sialidases that liberate glycan-linked Neu5Ac, and free Neu5Ac is an important growth substrate for certain commensal gut bacteria and pathogens (23–28). Because Neu5Ac released by extracellular sialidases is cross-fed between different bacterial species, Neu5Ac plays important ecological roles in the gut environment (25–27, 29, 30). In fact, Neu5Ac cross-feeding enables the expansion of opportunistic pathogens in mouse models of colitis (26, 27).

Interestingly, *R. gnavus* cleaves sialic acid by a mechanism that is distinct from well-known mucin-degrading gut bacteria, namely, by releasing 2,7-anhydro-Neu5Ac, rather than free Neu5Ac, as the cleavage product of glycan-linked Neu5Ac (31). This reaction is catalyzed by the NanH enzyme, an intramolecular *trans* (IT)-sialidase that specifically cleaves α 2,3-linked sialic acid (31). Notably, IT domains were reported to be enriched in the metagenomes of individuals with IBD compared to healthy controls (31). *R. gnavus* is able to use 2,7-anhydro-Neu5Ac as a growth substrate by virtue of an ABC family transporter that selectively imports 2,7-anhydro-Neu5Ac (but not Neu5Ac) into the cell (24) and an oxidoreductase NanOx that converts internalized 2,7-anhydro-Neu5Ac into Neu5Ac (24), which is further metabolized by the canonical Neu5Ac catabolism enzymes NanA, NanE, and NanK (Fig. 1A). The ABC transporter and *nan* genes are found within a single cluster in the *R. gnavus* genome (24). However, this gene cluster is not found in all *R. gnavus* strains, and the ability to utilize sialic acid and grow on mucin has been reported to vary by strain (32, 33).

Notably, an *R. gnavus* strain incapable of 2,7-anhydro-Neu5Ac catabolism is outcompeted by an otherwise isogenic strain in a germ-free mouse model, suggesting that this metabolism may be important for *R. gnavus* colonization of the gut. In multi-species microbial communities, cleavage of sialic acid by *R. gnavus* to form 2,7-anhydro-Neu5Ac may effectively sequester sialic acid from competing Neu5Ac-catabolizers, while potentially promoting the growth of a smaller subset of bacteria (including strains of *Escherichia coli* and *Salmonella typhimurium*) that also encode the 2,7-anhydro-Neu5Ac catabolism pathway (34). The aforementioned studies implicating *R. gnavus* sialic acid metabolism in disease, together with the ecological implications of the unique mechanism of sialic acid cleavage by this organism, prompted us to look more specifically into the distribution of sialic acid catabolism across diverse clinical isolates of *R. gnavus*.

By analyzing 77 *R. gnavus* genomes, including 17 newly sequenced in this study, from healthy people as well as individuals with IBD, we were able to profile the distribution of sialic acid catabolism within the *R. gnavus* species and discovered previously unknown variation in sialic acid utilization. Our findings contribute to a growing body of research

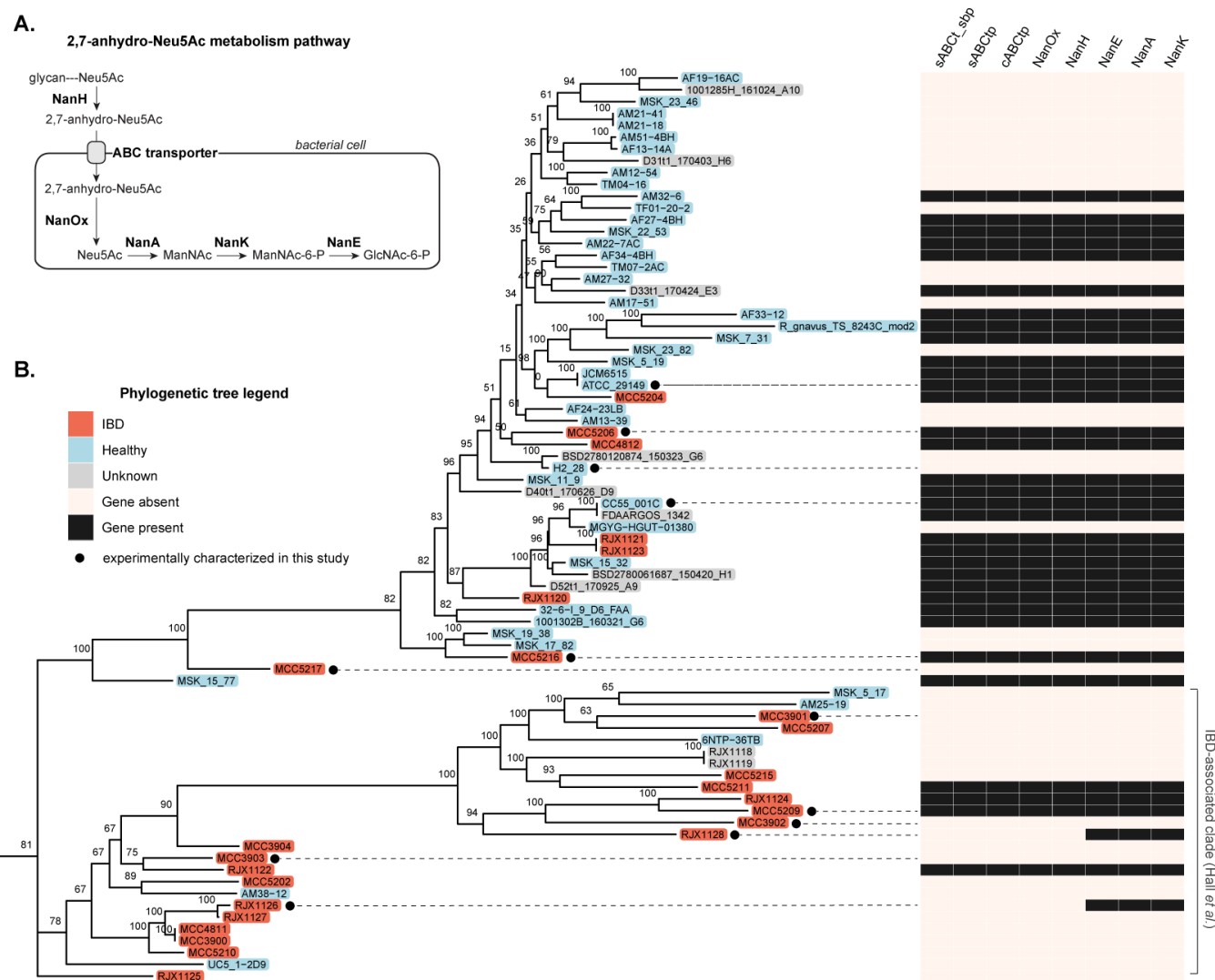

**FIG 1** (A) Proteins involved in 2,7-anhydro-Neu5Ac catabolism in *R. gnavus*. Abbreviated metabolites: ManNAc, *N*-acetylmannosamine; ManNAc-6-P, *N*-acetyl-mannosamine-6-phosphate; GlcNAc-6-P, *N*-acetylglucosamine-6-phosphate. (B) *R. gnavus* strain-level phylogeny and evidence for Nan protein-gene content for 77 sequenced *R. gnavus* genomes. The phylogenetic tree is based on a maximum-likelihood analysis of a 1,415,350 position core gene alignment and is unrooted. Branch labels indicate bootstrap support from 500 replicates. Strain names are colored based on their origins. Shown on the right is a genome-wide BLAST comparison of Nan protein-coding sequences. Filled boxes indicate high-scoring segment pairs (HSPs) yielded by BLASTp searches of putative protein sequences for the corresponding query *R. gnavus* genome against Nan protein sequences from reference strain *R. gnavus* ATCC 29149 with default parameter settings. Only HSPs satisfying >80% sequence identity and >60% sequence coverage are shown. Protein annotations: sABCt_sbp, sugar ABC transporter substrate-binding protein; sABCtp, sugar ABC transporter permease; cABCtp, carbohydrate ABC transporter permease.

emphasizing the essentiality of considering specific strains in future investigations into the roles of *R. gnavus* sialic acid metabolism in the gut (1).

## RESULTS

### Sialic acid metabolism is sporadically distributed among *R. gnavus* strains

The prevalence of 2,7-anhydro-Neu5Ac catabolism within the *R. gnavus* clade was investigated by comparing protein sequences from 77 *R. gnavus* genomes to Nan protein sequences from reference strain ATCC 29149 (24). The 77 genomes comprise 60 that were publicly available at the time the analysis commenced and a further 17 that were sequenced from an in-house collection, to which we refer as "MCC" strains. We included a significant proportion of strains from healthy individuals (41 of 77) to address

a limitation of previous strain-level genomic analyses of *R. gnavus*, which have predominantly focused on isolates from patients with IBD. To investigate the finding by Hall et al. that a sialic acid transporter may be constrained to a particular clade of *R. gnavus* (2), we constructed a core genome phylogenetic tree of *R. gnavus* strains and indicated the putative presence or absence of the 2,7-anhydro-Neu5Ac catabolism proteins across the strain tree (Fig. 1B).

Like Hall et al., we observed two clades of *R. gnavus*, with one clade showing a higher proportion of, but not exclusively composed of, IBD-associated strains (Fig. 1B; IBD-associated clade is marked on the far right). We found that roughly half (35/77) of the genomes encoded protein sequences with homology to Nan proteins, including strains isolated from healthy individuals (17/41 with Nan proteins) and from patients with IBD (11/26 with Nan proteins). All but two of these genomes yielded BLAST high-scoring segment pairs (HSPs) for all of the proteins in the 2,7-anhydro-Neu5Ac catabolism pathway, namely, the three-component ABC transporter (a substrate-binding protein and two transporter permeases), NanH, NanOx, NanE, NanA, and NanK (Fig. 1B). Among four divergent strains, we found ≥98.9% sequence identity in the Nan proteins at the amino acid level and apparent synteny of the *nan* gene cluster (Fig. S1). Strains RJX1126 and RJX1128 had "partial" pathways, yielding HSPs for NanE, NanA, and NanK only (Fig. 1B), which we examine in further detail in a following section. Overall, our analysis revealed that the 2,7-anhydro-Neu5Ac catabolism pathway is sporadically distributed throughout phylogenetically diverse strains. Thus, the sialic acid transporter that was identified by Hall et al. as belonging exclusively to an IBD-associated *R. gnavus* clade is unlikely to be the ABC family sialic acid transporter previously reported in this species.

## Sialic acid-dependent growth is consistent with the presence or absence of Nan proteins

To determine how the presence of Nan proteins corresponds to sialic acid utilization by live bacterial cells, we selected a dozen strains that span the phylogenetic diversity of this species (marked in Fig. 1B and listed in Table S1) and cultured them in basal YCFA medium supplemented with 2,7-anhydro-Neu5Ac, Neu5Ac, and 3'-sialyllactose, a human milk oligosaccharide consisting of sialic acid linked to lactose. Like *R. gnavus* ATCC 29149, all strains whose genomes encode all five Nan proteins and an ABC family transporter grew with 2,7-anhydro-Neu5Ac and 3'-sialyllactose, but not Neu5Ac (Fig. 2A). The absence of growth of all but one of these strains with lactose (Fig. S2) provides evidence that the strains use the sialic acid component of 3'-sialyllactose as a growth substrate. No major differences in the growth rate were observed between strains grown with 2,7-anhydro-Neu5Ac or 3'-sialyllactose (Fig. S3), suggesting that all the strains are efficient utilizers of these growth substrates. Strains lacking Nan proteins did not grow on any source of sialic acid (Fig. 2A).

Unlike the aforementioned strains, RJX1126 and RJX1128 grew to a high density with Neu5Ac (Fig. 2A). This phenotype is consistent with the presence of NanA, NanK, and NanE enzymes in the genomes of these strains. The inability of strains RJX1126 and RJX1128 to grow with 3'-sialyllactose is also consistent with the absence of a genomically encoded sialidase. Both strains grew to an appreciable density with 2,7-anhydro-Neu5Ac, which was surprising given that they lack the 2,7-anhydro-Neu5Ac importer and NanOx. To test for Neu5Ac contamination of our 2,7-anhydro-Neu5Ac stock, we performed two-dimensional NMR analysis of the enzymatically synthesized 2,7-anhydro-Neu5Ac; however, we did not observe Neu5Ac by this method (Fig. S4). The apparent stability of 2,7-anhydro-Neu5Ac in bYCFA over the course of 24 hours (Fig. S5) suggests that growth was also not due to spontaneous conversion of 2,7-anhydro-Neu5Ac to Neu5Ac in the growth medium. Still, trace quantities of Neu5Ac, or an indirect effect of bacterial growth on 2,7-anhydro-Neu5Ac stability in culture medium, could explain this result.

Finally, we screened all 12 *R. gnavus* clinical isolates for the ability to grow with purified porcine gastric mucin (pPGM). As previously reported, strain ATCC 29149 reached a moderate density with pPGM as the sole added carbon source (32). Many of

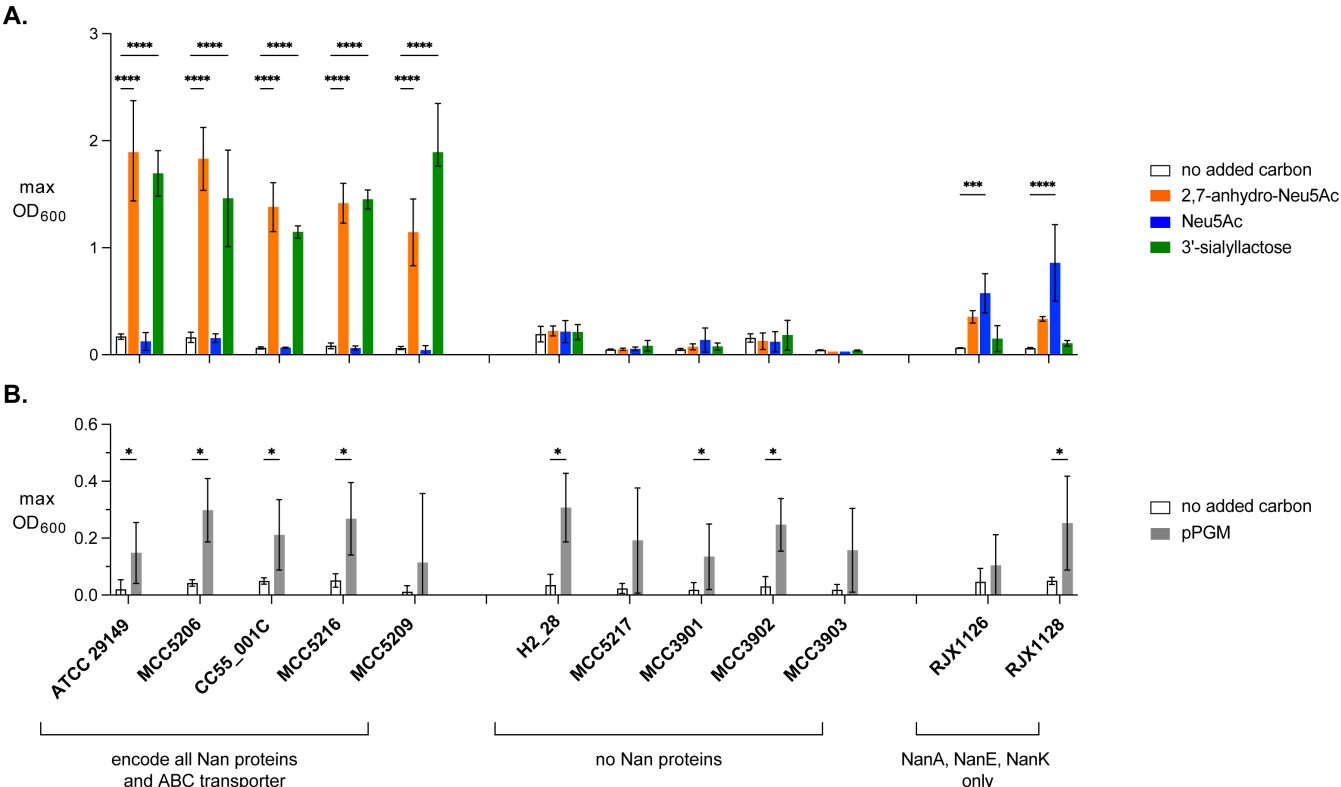

**FIG 2** Growth of *R. gnavus* strains with various carbon sources. Maximum optical densities reached by strains in culture during 24 hours of growth with (A) simple sugars or (B) purified porcine gastric mucin (pPGM). Bars and brackets indicate the average and standard deviations, respectively, of at least three independent experiments. (A) $n = 3$, (B) $n = 8$ biological replicates. Significance was assessed using (A) a two-way ANOVA with Dunnett's multiple comparisons tests or (B) multiple paired *t*-tests with a 1% false discovery rate threshold; *q < 0.01, ***P < 0.001; ****P < 0.0001.

the other *R. gnavus* isolates grew with pPGM, including strains that lack sialic acid catabolism (Fig. 2B; Fig. S6), suggesting that the ability to cleave and metabolize sialic acid is not a major determinant of growth with pPGM. Indeed, previously published studies demonstrate that other glycoside hydrolases expressed by *R. gnavus*, namely, fucosidases and endo-galactosidases, enable utilization of pPGM as a growth substrate (32, 35, 36).

## The Nan protein cluster in *R. gnavus* strains RJX1126 and RJX1128 functions in Neu5Ac catabolism

The NanA, NanE, and NanK proteins found in *R. gnavus* strains RJX1126 and RJX1128 appear to be distinct homologs from those found in other *R. gnavus* strains; these proteins share 91.5%, 86.96%, and 80.82% amino acid identity, respectively, with the ATCC 29149 homologs, whereas the NanA, NanE, and NanK sequences in all other strains are more highly conserved (Fig. S1). Furthermore, the *nanAEK* genes in *R. gnavus* strains RJX1126 and RJX1128 are found within a distinct gene cluster (Fig. 3A and Table 1) that lacks the 2,7-anhydro-Neu5Ac metabolism-specific genes *nanH* and *nanOx* and lacks the ABC family 2,7-anhydro-Neu5Ac importer. Instead, the *nan* genes in these two strains are clustered with a sodium:solute symporter (SSS) gene. SSS family transporters are one of four classes of transporters involved in the import of Neu5Ac (37, 38), and the SSS protein in *R. gnavus* shares 68.6% identity with a verified Neu5Ac transporter protein in *Clostridioides (Clostridium) difficile* (26). Interestingly, this locus also encodes proteins putatively involved in transposition and reverse transcription, suggesting it may have been horizontally acquired.

The genetic content of the *nan* cluster in *R. gnavus* strains RJX1126 and RJX1128 is consistent with the ability of these two strains to utilize Neu5Ac as a growth substrate

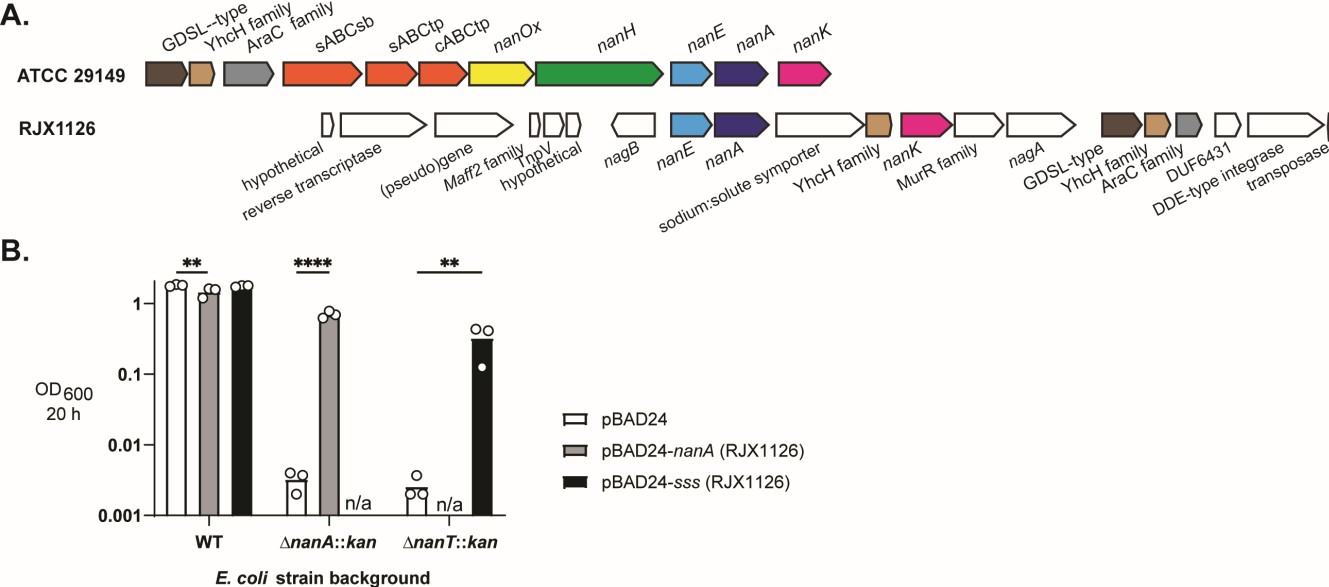

**FIG 3** (A) Gene neighborhood of the *nan* genes in *R. gnavus* strain RJX1126, contrasted to the previously described *nan* cluster in strain ATCC 29149. (B) Complementation of *E. coli* BW25113 *nanA* and *nanT* (sialic acid transporter) mutants with *R. gnavus* RJX1126 *nanA* and sodium:solute symporter (*sss*) genes, tested by growth in the M9 medium with Neu5Ac as the sole carbon source. Data points show the density achieved by each strain after 20 h of growth, in three independent experiments; columns indicate mean $OD_{600}$ values. "n/a": this condition was not tested. Significance was assessed using a two-way ANOVA with Dunnett's multiple comparisons tests; **$P < 0.01$; ****$P < 0.0001$.

(Fig. 2A). To probe the hypothesis that the distinct gene cluster we discovered in these strains functions in Neu5Ac catabolism, we tested whether the RJX1126 *nanA* and SSS-encoding genes would complement mutations in the Neu5Ac catabolism pathway of the laboratory strain *Escherichia coli* BW25113. The RJX1126 *nanA* gene restored Neu5Ac-dependent growth of *E. coli* containing a *nanA* disruption, and the SSS gene restored the growth of an *E. coli nanT* (Neu5Ac transporter of the major facilitator

**TABLE 1** Putative sialic acid catabolism gene cluster in *R. gnavus* strain RJX1126 (genome accession number NZ_NIHU01000030.1)

| Locus tag | Genomic location | Current NCBI annotation | Comments |
|---|---|---|---|
| CDL24_RS14640 | 1–132 | Transposase domain-containing protein | |
| CDL24_RS14645 | 178–1,497 | DDE-type integrase/transposes/recombinase | |
| CDL24_RS14650 | 1,571–2,041 | DUF6431 domain-containing | |
| CDL24_RS14655 | 2,242–2,697 | AraC family ligand-binding domain-containing protein | Transcriptional regulator |
| CDL24_RS14660 | 2,768–3,223 | YhcH/YjgK/YiaL family protein | Possible sugar anomerase |
| CDL24_RS14665 | 3,244–3,954 | GDSL-type esterase/lipase family protein | Possible acyl hydrolase |
| CDL24_RS14670 | 4,388–5,542 | N-acetylglucosamine-6-phosphate deacetylase | *nagA* |
| CDL24_RS14675 | 5,587–6,426 | MurR/PriR family transcriptional regulator | |
| CDL24_RS14680 | 6,445–7,332 | ROK family protein | Putative *nanK* |
| CDL24_RS14685 | 7,467–7,919 | YhcH/YjgK/YiaL family protein | Possible sugar anomerase |
| CDL24_RS14690 | 7,935–9,440 | Sodium:solute symporter | Putative Neu5Ac transporter |
| CDL24_RS14695 | 9,546–10,463 | Dihydrodipicolinate synthase family protein | Putative *nanA* |
| CDL24_RS14700 | 10,518–11,207 | N-acetylmannosamine-6-phosphate 2-epimerase | *nanE* |
| CDL24_RS14705 | 11,488–12,216 | Glucosamine-6-phosphate deaminase | *nagB* |
| CDL24_RS14715 | 12,710–12,991 | Hypothetical protein | |
| CDL24_RS14720 | 13,006–13,353 | TnpV protein | Transposon-encoded protein |
| CDL24_RS14725 | 13,427–13,588 | Maff2 family protein | Short transmembrane protein |
| CDL24_RS14730 | 13,865–15,187 | Uncharacterized (pseudo)gene | |
| CDL24_RS14735 | 15,297–16,787 | Reverse transcriptase domain-containing protein | |
| CDL24_RS15960 | 16,903–17,103 | Hypothetical protein | |

superfamily) mutant, demonstrating that these *R. gnavus* RJX1126 genes indeed function in Neu5Ac import and catabolism. The presence of the SSS in *R. gnavus* strains RJX1126 and RJX1128 likely confers their unique (within the species *R. gnavus*) ability to grow with Neu5Ac, as the SSS protein is absent from all other *R. gnavus* genomes examined in this study.

## DISCUSSION

The gut commensal species *R. gnavus* catabolizes sialic acid by a mechanism that differs from most sialic acid-metabolizing organisms (31). Because of the known roles of sialic acid catabolism in mucus homeostasis and microbial ecology (17, 26, 27, 29), and due to multiple reports that *R. gnavus* is positively associated with gut inflammatory disease (2–12), the biological impacts of the 2,7-anhydro-Neu5Ac metabolism of this organism were of interest to us. However, sialic acid and mucus metabolism are known to vary between *R. gnavus* strains (31–33). Here, we analyzed the prevalence and variation of sialic acid catabolism proteins across 77 strains, finding that 2,7-anhydro-Neu5Ac catabolism was sporadically distributed among strains isolated from both healthy individuals and patients with IBD and discovering further diversity than was previously appreciated. Based on our results, we can classify *R. gnavus* strains into three categories of sialic acid metabolism: strains that cleave glycan-linked sialic acid to generate 2,7-anhydro-Neu5Ac and use this substrate for growth; strains that require free Neu5Ac for sialic acid-dependent growth; and strains that do not utilize sialic acid as a growth substrate (Fig. 4).

As mentioned in describing the motivation for this study, Hall et al. identified a sialic acid transporter that was absent from isolates from healthy individuals and restricted to an IBD-specific clade (2). Based on our examination of the distribution of sialic acid catabolism across *R. gnavus* strains, we conclude that the sialic acid transporter that emerged from their study was the Neu5Ac SSS family transporter that we identified in strains RJX1126 and RJX1128. The potential relevance of Neu5Ac catabolism for *R. gnavus* growth in inflammatory conditions is of interest given that high free Neu5Ac levels are associated with dysbiosis (26, 27). Curiously, strains RJX1126 and RJX1128 also grew with 2,7-anhydro-Neu5Ac as the sole carbon source, albeit to a significantly lower density compared to strains that have the complete *nan* cluster. As the genomes of strains RJX1126 and RJX1128 lack the gene encoding NanOx, we suspect that their growth on 2,7-anhydro-Neu5Ac was due to spontaneous conversion of a fraction of this substrate to Neu5Ac; however, we were unable to experimentally validate this hypothesis. A recent report that two bacterial sialic acid transporters of the SSS family could not import 2,7-anhydro-Neu5Ac further supports our conclusion that the growth of *R. gnavus* strains RJX1126 and RXJ1128 was not by direct utilization of 2,7-anhydro-Neu5Ac (38).

Despite variations in sialic acid catabolism among the *R. gnavus* strains we tested, all but one of the strains grew with pPGM as the sole carbon source. Importantly, this finding may or may not translate to growth with human intestinal mucin, which is expected to differ significantly in glycan composition and sugar linkages compared to PGM. Indeed, previous work on the established mucin degrader *A. muciniphila* demonstrates that growth with oligosaccharides derived from different mucin sources can vary drastically based on the glycan structure (41). However, taken at face value, the finding that almost all of the *R. gnavus* strains tested grew with mucin suggests that different strains have evolved different mechanisms to utilize this important nutrient source. In addition to the sialidase gene, fucosidase and endo-galactosidase genes were upregulated in *R. gnavus* strains ATCC 29149 and ATCC 35913 when they were grown with mucin (32, 33). Interestingly, endo-β−1,4-galactosidase activity was later demonstrated to enable *R. gnavus* strain E1, which cannot otherwise grow with pPGM, to use mucin as a growth substrate (35). Likewise, a specific fucosidase activity may enable *R. gnavus* strains to catabolize sialylated substrates without removing sialic acid (36). Future studies comparing the mechanisms of growth of the *R. gnavus* strain examined in this study with pPGM would elucidate the extent of diversity of mucin degradation strategies within this species.

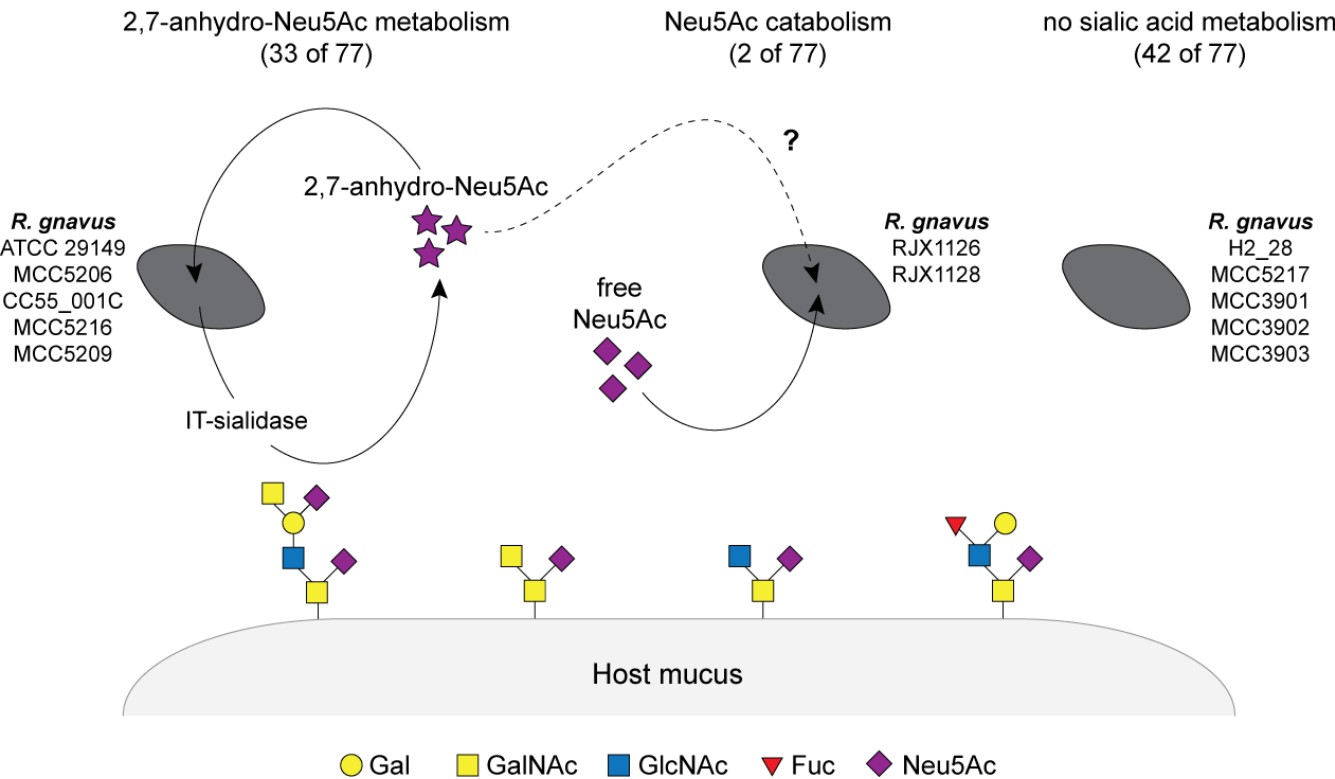

**FIG 4** Variable interactions of *R. gnavus* strains with mucus-derived sialic acids. Left: strains that cleave glycan-linked Neu5Ac to generate 2,7-anhydro-Neu5Ac and use 2,7-anhydro-Neu5Ac as a growth substrate. Middle: strains that grow with free Neu5Ac (and to a lesser extent with free 2,7-anhydro-Neu5Ac, although this phenotype requires further investigation). Right: strains that do not catabolize sialic acid. The numbers above each cartoon indicate the fraction of analyzed *R. gnavus* genomes that fall into each category of sialic acid metabolism; strains that were experimentally characterized in this work are listed in their corresponding category. Glycans are depicted according to symbol nomenclature for glycan systems and represent abundant sialylated oligosaccharides in the human colonic mucin MUC2 (39, 40) (Gal, galactose; GalNAc, *N*-acetylgalactosamine; GlcNAc, *N*-acetylglucosamine; Fuc, fucose).

The biological impact of differential sialic acid catabolism in *R. gnavus* is an exciting topic for further exploration. The finding that *R. gnavus* strains have evolved different ways to utilize sialic acid suggests the importance of this nutrient source in the gut environment. Based on the distinct sialic acid catabolism pathways presented in Fig. 4, we hypothesize that *R. gnavus* strains would differentially respond to changes in mucus (e.g., quantity of mucus and degree of mucin sialylation) and would be differentially impacted by levels of general sialidase activity, and consequently different levels of free sialic acid, which are known to be altered in healthy versus diseased states of the gut (26, 27). Additionally, we would predict distinct ecological interactions of *R. gnavus* with other gut bacteria based on sialic acid catabolism. For example, *R. gnavus* strains that generate 2,7-anhydro-Neu5Ac may be involved in cross-feeding interactions with other 2,7-anhydro-Neu5Ac-metabolizing bacteria and may sequester sialic acid from bacteria that typically take advantage of Neu5Ac. In contrast, *R. gnavus* strains that can utilize Neu5Ac as a growth substrate might benefit from the presence of gut bacterial species that are Neu5Ac producers. These types of interactions may contribute to changes in microbial community composition that occur in gut dysbiosis, which is a critical component of IBD (42). Thus, examining specific and relevant *R. gnavus* strains will be key to untangling potential roles of the sialic acid metabolism of this species in gut biology in health and disease.

## MATERIALS AND METHODS

### Materials

2,7-anhydro-Neu5Ac was prepared from 2′-(4-methylumbelliferyl)-*α*-D-*N*-acetylneuraminic acid (4-MU-Neu5Ac) purchased from Carbosynth (see "Enzymatic synthesis of 2,7-anhydro-Neu5Ac"). *N*-acetylneuraminic acid was purchased from Sigma and Carbosynth. 3′-sialyllactose was purchased from Carbosynth. Porcine gastric mucin (type III) was purchased from Sigma and further purified (see "Mucin preparation"). Sources of *R. gnavus* strains are detailed in Table S1 (2, 43–46).

### Strain isolation and genome sequencing

*R. gnavus* strains were isolated from stool samples from participants of phase 3 trials for the treatment of Crohn's disease and ulcerative colitis (45, 46). Isolations were performed by Microbiotica Ltd. using a proprietary platform that utilizes sequencing- and culturing-guided strain selection (47). Genomic DNA was extracted using the Qiagen DNeasy UltraClean Microbial Kit. Quality of the genomic DNA was assessed using the Genomic DNA ScreenTape and TapeStation 4200 (Agilent Technologies). Genomic DNA was quantified using the Qubit dsDNA BR assay kit (Thermo Fisher Scientific). For library preparation, the Nextera DNA Flex kit (Illumina) was used with an input of 100 ng of genomic DNA. The resulting libraries were multiplexed and sequenced on NovaSeq (Illumina) to generate 5 million paired-end 75 base pair reads for each sample. Genomes were assembled and annotated with Spades v3.13.1 (48) and Prokka v1.12 (49), respectively, using default settings. Assembly metrics are detailed in Table S3. Genome quality was assessed using CheckM v1.2.3 (50) (Table S4).

### Phylogenetic tree construction and annotation

An *R. gnavus* phylogenetic tree was constructed using 60 publicly available genomes downloaded from the NCBI GenBank database and 17 genomes sequenced in this study (see "Strain isolation and genome sequencing"). A list of the genome sequences used in this analysis can be found in Table S2. Roary (51) was used to construct a core gene alignment (present in 95% of isolates), from which a phylogenetic tree was estimated with RAxML v8 (52) using the generalized time-reversible (GTR) model with gamma correction for among-site rate variation. Support for nodes was assessed using 500 bootstrap replicates. The tree is unrooted. R packages tidyverse, ggtree, ape, and treeio were used for visualization. Strain origins were annotated by referencing source studies. Genome protein sequences were compared by BLASTp v2.2.28 (53) to Nan and 2,7-anhydro-Neu5Ac catabolism protein sequences from *R. gnavus* ATCC 29149 (sugar ABC transporter substrate-binding protein, RGna_RS08355; sugar ABC transporter permease, RGna_RS08350; carbohydrate ABC transporter permease, RGna_RS08345; NanOx, RGna_RS08340; NanH, RGna_RS08335; NanE, RGna_RS08330; NanA, RGna_RS08325; NanK, RGna_RS08320) (24). A protein was considered present if BLAST searches yielded HSPs satisfying >80% identity and >60% sequence coverage.

### Bacterial culturing

*R. gnavus* was cultured inside of a Coy anaerobic chamber, with a headspace of 85% $N_2$/5% $H_2$/10% $CO_2$, in basal YCFA (32, 54) (bYCFA; containing, per liter: 10 g casitone, 2.5 g yeast extract, 4 g $NaHCO_3$, 1 g L-cysteine, 450 mg $K_2HPO_4$, 450 mg $KH_2PO_4$, 900 mg $(NH_4)_2SO_4$, 900 mg NaCl, 90 mg $MgSO_4$, 90 mg $CaCl_2$, 1 mg resazurin, 10 mg hemin, 10 µg biotin, 10 µg cyanocobalamin, 30 µg *p*-aminobenzoic acid, 50 µg folic acid, 150 µg pyridoxamine, 4.05 mL acetate, 1.43 mL propionate, 0.22 mL n-valerate, isovalerate and isobutyrate, pH 6.5; and 50 µg thiamine, and 50 µg riboflavin, added after autoclaving) supplemented with 0.2% glucose, at 37°C. For testing growth on various carbon sources, *R. gnavus* strains were pre-cultured overnight in bYCFA with 0.1% glucose and diluted 1:200 into fresh bYCFA containing the specified growth substrate at a concentration

of 11.1 mM, or 1% wt/vol mucin (24, 32). Growth rates were measured by culturing in 96-well plates in an Epoch2 plate reader (BioTek), at 37°C, with continuous orbital shaking.

## Heterologous expression and purification of NanH for 2,7-anhydro-Neu5Ac synthesis

The *R. gnavus* ATCC 29149 *nanH* gene [truncated to remove the N-terminal signaling sequence (31)] was codon-optimized for *E. coli* and cloned with an N-terminal hexa-histidine tag into the pET11a vector (see Supplementary Materials and Methods) for expression in *E. coli* BL21-Gold(DE3). The expression strain carrying the pET11a-*nanH* plasmid was pre-cultured at 37°C in LB medium with carbenicillin (50 ug/mL) until $OD_{600}$ 0.8–0.9, after which protein expression was induced at 16°C for 24 hours by addition of 1 mM isopropyl β-D-1-thiogalactopyranoside. Frozen bacterial cell pellets were resuspended in 50 mM Tris-HCl, 150 mM sodium chloride, pH 7.8, supplemented with 10 mM imidazole, cOmplete Mini EDTA-free Protease Inhibitor Cocktail, and 0.1 mg/mL DNase, and lysed with BugBuster Protein Extraction Reagent (Millipore). The protein was batch-bound to Ni-NTA resin, washed with 50 mM Tris-HCl, 150 mM sodium chloride, pH 7.8, 10 mM imidazole, and eluted using an increasing concentration of imidazole in the same buffer. Eluate purity was analyzed by SDS-PAGE using Coomassie stain, and fractions enriched in NanH were concentrated and exchanged into 10 mM HEPES, pH 8.0, with 10% glycerol by repeated centrifugation in Amicon 30 kDa molecular weight cut-off (MWCO) spin filters. Purified protein was flash-frozen with liquid nitrogen and stored at −80°C.

## Enzymatic synthesis of 2,7-anhydro-Neu5Ac

2,7-anhydro-Neu5Ac was prepared as previously described (31), by treatment of 4-MU-Neu5Ac (100 mM) with the *R. gnavus* NanH enzyme (0.01 mg/mL) in 20 mM $Na_2HPO_4$, 20 mM $NaH_2PO_4$, pH 6.5, containing 1 mg/mL bovine serum albumin, at 37°C. The progression of the reaction was monitored fluorimetrically (excitation spectra 340 nm; emission spectra 420 nm). Upon completion of the reaction, the leaving group 4-methylumbelliferone crystallized out of solution and was removed. The reaction was then quenched by addition of ethanol (50% vol/vol), and the precipitate was removed by filtration. An Amicon 10 MWCO spin filter was used to remove the residual protein, and product purity was evaluated by liquid chromatography-mass spectrometry (LC-MS) and 1D/2D $^{1}H$ NMR (see Supplementary Materials and Methods). 2,7-anhydro-Neu5Ac was dried, quantified using quantitative NMR (with caffeine as an external standard), dissolved in water at a stock concentration of 222 mM, and stored at −20°C.

## Mucin preparation

Mucin was prepared following previously established protocols (25, 55, 56). Commercially available porcine gastric was dissolved in Dulbecco's phosphate buffered saline, pH 7.4, at 2.5% wt/vol. After 1–2 hours stirring at room temperature, the pH was adjusted to 7.4, and the suspension was stirred overnight. Insoluble debris was removed by centrifugation (10,000 x *g*, 4 degrees, 30 minutes). Mucin was further purified and sterilized by addition of ice-cold ethanol (60% vol/vol, final). The precipitated mucin was collected by gentle centrifugation (250 x *g*, 1 minute) and resuspended in water. To remove residual salts, the mucin solution was dialyzed against water using a 10 MWCO Spin-A-Lyzer G2 dialysis cassette (Pierce Biotechnology). Purified mucin was lyophilized and stored at −20°C.

## Genetic complementation assays

Predicted *nanA* and sodium:solute symporter (*sss*) open reading frames (contig NIHU0100030.1, nucleotides 7000..7917 and 8023..9528, respectively) were amplified from *R. gnavus* strain RJX1126 and cloned into pBAD24 at the NcoI restriction

site. Plasmids were electroporated into *E. coli* Keio collection strains (57) (wild-type, Δ*nanA::kan*, and Δ*nanT::kan*). Transformed strains were grown in M9 medium with 0.2% glucose and 50 µg/mL carbenicillin and transferred to M9 medium containing 11.1 mM Neu5Ac and 50 µg/mL carbenicillin for growth assays. Because of the leaky expression of the arabinose-inducible promoter, complementation was achieved in the absence of an inducer (addition of arabinose was toxic to strains expressing *R. gnavus nanA* or *sss*; data not shown). Growth kinetics were measured by culturing in a 96-well plate in a BioTek Synergy2 plate reader at 37°C with continuous shaking.

## ACKNOWLEDGMENTS

We would like to thank Fiona Tamburini for generating early versions of the *R. gnavus* strain-level phylogenetic tree; Genentech colleagues and Kristopher Kennedy for many helpful discussions about this work; and Dr. Ramnik Xavier and his laboratory members for generously sharing strains RJX1126 and RJX1128.

All authors are Genentech/Roche employees and may own Roche stock. The authors have no other relevant affiliations or financial involvement with any organization or entity with a financial interest in or financial conflict with the subject matter or materials discussed in the manuscript apart from those disclosed.

## AUTHOR AFFILIATIONS

[1]Department of Infectious Diseases & Host-Microbe Interactions, Genentech, Inc., South San Francisco, California, USA

[2]Department of OMNI Bioinformatics, Genentech Inc., South San Francisco, California, USA

[3]Department of Discovery Chemistry, Genentech, Inc., South San Francisco, California, USA

[4]Department of Drug Metabolism and Pharmacokinetics, Genentech, Inc., South San Francisco, California, USA

## PRESENT ADDRESS

Olga M. Sokolovskaya, Department of Biological Engineering, Massachusetts Institute of Technology, Cambridge, Massachusetts, USA

Jasmina Uzunovic, Department of DSS Hematology, Roche, Canada

## AUTHOR ORCIDs

Olga M. Sokolovskaya http://orcid.org/0000-0003-1202-4248

Yutian Peng http://orcid.org/0000-0002-8663-7330

## DATA AVAILABILITY

Sequencing data are available from NCBI under BioProject ID PRJNA1283276 with BioSample accessions SAMN49687146, SAMN49687147, SAMN49687148, SAMN49687149, SAMN49687150, SAMN49687151, SAMN49687152, SAMN49687153, SAMN49687154, SAMN49687155, SAMN49687156, SAMN49687157, SAMN49687158, SAMN49687159, SAMN49687160, and SAMN49687161.

## ADDITIONAL FILES

The following material is available online.

### Supplemental Material

**Supplemental materials (Spectrum03090-24-s0001.docx).** Fig. S1 to S6, Tables S1 to S4, supplemental materials and methods.

Open Peer Review

**PEER REVIEW HISTORY (review-history.pdf).** An accounting of the reviewer comments and feedback.

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
