## [Reviewer comments · Microbiology Spectrum]

Microbiology Spectrum

Dysbiosis-associated gut bacterium *Ruminococcus gnavus* varies at the strain level in ability to utilize key mucin component sialic acid

Olga Sokolovskaya, Jasmina Uzunovic, Yutian Peng, Mikiko Okumura, Lingjue Wang, Yuhui Zhou, Zijuan Lai, Elizabeth Skippington, and Man-Wah Tan

Corresponding Author(s): Olga Sokolovskaya, Genentech Inc

Review Timeline:

Submission Date:	January 28, 2025
Editorial Decision:	February 19, 2025
Revision Received:	May 5, 2025
Accepted:	May 14, 2025

Editor: Jennifer Auchtung

Reviewer(s): The reviewers have opted to remain anonymous.

Transaction Report:

DOI: <https://doi.org/10.1128/spectrum.03090-24>

Re: Spectrum03090-24 (Dysbiosis-associated gut bacterium *Ruminococcus gnavus* varies at the strain level in ability to utilize key mucin component sialic acid)

Dear Dr. Olga Matveevna Sokolovskaya:

Thank you for the privilege of reviewing your work. I appreciated your efforts to address the majority of previous reviewer comments. However, I agree with your previous reviewers that you must provide statistical analyses for Figures 2A and 2B to meet the expected standards for publication within this field and make this manuscript potentially acceptable for publication.

Revision Guidelines

Sincerely,
Jennifer Auchtung
Editor
Microbiology Spectrum

Re: Spectrum03090-24R1 (Dysbiosis-associated gut bacterium *Ruminococcus gnavus* varies at the strain level in ability to utilize key mucin component sialic acid)

Dear Dr. Olga Matveevna Sokolovskaya:

Your manuscript has been accepted, and I am forwarding it to the ASM production staff for publication. Your paper will first be checked to make sure all elements meet the technical requirements. ASM staff will contact you if anything needs to be revised before copyediting and production can begin. Otherwise, you will be notified when your proofs are ready to be viewed.

Sincerely,
Jennifer Auchtung
Editor
Microbiology Spectrum